# Pre-Analytical Factors Affecting Extracellular DNA in Saliva

**DOI:** 10.3390/diagnostics14030249

**Published:** 2024-01-24

**Authors:** Ľubica Janovičová, Dominika Holániová, Barbora Vlková, Peter Celec

**Affiliations:** 1Institute of Molecular Biomedicine, Faculty of Medicine, Comenius University, 811 08 Bratislava, Slovakia; lubica.janovicova@gmail.com (Ľ.J.); dholaniova@gmail.com (D.H.); barboravlk@gmail.com (B.V.); 2Institute of Pathophysiology, Faculty of Medicine, Comenius University, 811 08 Bratislava, Slovakia

**Keywords:** cell-free DNA, sputum, periodontitis, neutrophil extracellular traps, microparticles

## Abstract

Salivary DNA is widely used for genetic analyses because of its easy collection. However, its extracellular fraction in particular, similar to the extracellular DNA (ecDNA) in plasma, could be a promising biomarker for oral or systemic diseases. In contrast to genetics, the quantity of salivary ecDNA is of importance and can be affected by the pre-analytical processing of samples, but the details are not known. The aim of our study was to analyze the effects of centrifugation and freezing of saliva on the concentration of ecDNA in saliva. Fifteen healthy volunteers, free of any known systemic or oral diseases, were asked to collect unstimulated saliva samples. Aliquots were centrifuged at 1600× *g* and frozen or directly processed. The fresh or thawed cell-free saliva samples underwent subsequent centrifugation at 16,000× *g*. The supernatants were used for DNA isolation and quantification using fluorometry and real-time PCR. While freezing had minimal effects on the salivary ecDNA concentration, another centrifugation step decreased ecDNA considerably in both fresh and frozen samples (by 97.8% and 98.4%, respectively). This was mirrored in the quantitative PCR targeting a nuclear (decrease by 93.5%) and mitochondrial (decrease by 97.7%) ecDNA sequence. In conclusion, in this first study focusing on the technical aspects of salivary ecDNA quantitation, we show that, regardless of its subcellular origin, the concentration of ecDNA in saliva is mainly affected by additional centrifugation and not by the freezing of centrifuged cell-free saliva samples. This suggests that most salivary ecDNA likely is associated with cell debris and apoptotic bodies. Which fraction is affected by a particular disease should be the focus of further targeted studies.

## 1. Introduction

DNA can exit the cell after cell damage and death or actively via the production of extracellular traps [1]. This extracellular DNA (ecDNA) or cell-free DNA is a valuable biomarker for non-invasive prenatal diagnosis or cancer progression monitoring [2]. Most analyses of ecDNA are conducted by sequencing, but in some contexts, more important than the sequence is the quantity of ecDNA, especially if the source of ecDNA are neutrophil extracellular traps [3]. The concentration of ecDNA can then provide information on the intensity of immune system activation.

EcDNA is widely studied in blood plasma, as blood is widely used for clinical biochemistry [4,5]. It is also clear that most studies on the biology of ecDNA are conducted with blood plasma [6]. However, ecDNA is also present in other body fluids. In the urine, it is analyzed in relation to kidney and bladder diseases [1] but can also be of systemic origin and reflect pregnancy or the status of a transplanted organ [7,8]. In cerebrospinal fluid, the most common application of ecDNA analysis is the detection or monitoring of brain tumors [9,10]. It is, nevertheless, not surprising that widely accepted consensus protocols for the processing of a sample and isolation of ecDNA exist only for blood plasma [11,12,13].

Saliva is a nearly ideal biological fluid. The sampling is easy, non-invasive, and cheap, but, most importantly, it does not require any intervention from a doctor or a nurse enabling the collection by the patient [14]. This presumably technical detail has broad implications. Cancer, but also other diseases, can be screened, and easy, cheap, and non-invasive screening is the key to real-life effectivity. Therefore, saliva and saliva-based biomarkers have great potential for clinical applications [15]. This is true for hormones, markers of oxidative stress, and other analytes, but mostly true with respect to salivary DNA.

DNA in saliva is widely used in genetic studies, especially if collection is to be done at home [16]. For these genetic studies, it is not important whether the DNA comes from epithelial cells, neutrophils, or blood cells due to blood contamination. It is important that the DNA comes in high concentration and is ideally not fragmented, nor contaminated with food [17,18]. Intracellular DNA usually fits this purpose if there are enough cells in saliva. Saliva, however, also contains ecDNA, which is fragmented and present in a much lower concentration [19,20].

Saliva has nuclease activity, but it is much lower than in plasma or urine [21]. The main determinant of ecDNA concentration in saliva seems to be its production. In patients with oral carcinoma, the concentration is unsurprisingly higher than in controls [22] and higher than in patients with benign precancerous lesions [23]. In addition, both the nuclear and the mitochondrial fraction of salivary ecDNA seem to be of importance and have some prognostic value [24]. This underlines the importance of studying the pre-analytical factors that could greatly affect the measured concentration of various types of ecDNA in saliva.

DNA found in saliva originates from the host tissues, but bacterial DNA is one of the additional contributors, apart from DNA, originating from food. In many cases, saliva is centrifuged to remove large debris consisting of cells and bits of food. However, smaller debris that is in plasma, called microparticles, is found in saliva too since microparticles and exosomes are released from activated and dying cells [25]. In plasma, this smaller debris contains DNA, RNA, and various proteins and can act as cargo to be delivered to other cells [26].

The aim of our study was to analyze the effects of the centrifugation and freezing of saliva samples from healthy volunteers on the concentration of total, nuclear, and mitochondrial ecDNA in saliva. In addition, we wanted to prove whether there was a correlation between various fractions of salivary ecDNA prepared for DNA isolation in various ways. Our hypothesis was that centrifugation would decrease and freezing would increase the ecDNA concentrations, especially of nuclear origin.

## 2. Methods

### 2.1. Saliva Collection and Sample Processing

Unstimulated saliva from 15 healthy volunteers, free of any known systemic or oral disease, based on the recent preventive check-up examinations was collected in 15 mL falcon tubes. The samples were divided into aliquots and centrifuged right after saliva collection at 1600× *g* for 10 min at 4 °C to remove cells. The supernatant was transferred into clean tubes and either centrifuged again at 16,000× *g* for 10 min at 4 °C or used for DNA isolation. Thereafter, ecDNA was isolated from saliva samples centrifuged once at 1600× *g* or centrifuged twice—at 1600× *g* and then at 16,000× *g*. In addition, two aliquots were frozen after the first centrifugation. One of them was centrifuged again after thawing at 16,000× *g*. Both were also used for DNA isolation. The procedure with the compared pre-analytical protocols for the processing of saliva was summarized in Figure 1. Salivary DNA from samples centrifuged at 1600× *g*, 16,000× *g* and microparticles found in pellets after centrifugation at 16,000× *g* was isolated and quantified. Additionally, one aliquot of saliva was used for DNase I treatment (1.5 K.u.) to assess the sensitivity to being cleaved. Each DNase I treated sample was incubated for 30 min at 37 °C. The study protocol was approved by the ethics committee of the Institute of Molecular Biomedicine. All the volunteers signed their informed consent.

### 2.2. Extracellular DNA Isolation and Quantification

EcDNA was isolated using QIAamp DNA Mini Kit on the QIAcube instrument (Qiagen, Hilden, Germany). Two hundred microliters of the corresponding saliva supernatants were used for the isolation of DNA according to the protocol from the manufacturer. DNA was eluted into 100 μL of Millipore water. Isolated ecDNA was used for subsequent quantitative analysis using fluorometry and real-time PCR.

The concentration of total ecDNA was determined using Qubit dsDNA high sensitivity assay kit (Thermo Fisher, Waltham, MA, USA) on a Qubit 3.0 fluorometer. Briefly, a reaction mixture was prepared based on the protocol from the manufacturer. Fluorescence was measured three times in thin-walled 0.5 mL tubes (Eppendorf, Hamburg, Germany). Standards of DNA for calibration were used for automatic quantitation. Average concentrations in ng/mL were used for subsequent analysis. Technical variability was below 3%.

Subcellular fractions of ecDNA were determined based on the real-time PCR targeted at single-copy sequences of the nuclear and mitochondrial genome. Primers for nuclear DNA (beta globin gene) were forward 5′-GCTTCTGACACAACTGTGTTCACTAGC-3′ and reverse 5′-CACCAACTTCATCCACGTTCACC-3′. Primers for mitochondrial DNA (D-loop) were forward 5′-CATAAAAACCCAATCCACATCA-3′ and reverse 5′-GAGGGGTGGCTTTGGAGT-3′. The reaction mixture contained SsoAdvanced universal SYBR Green supermix (Biorad, Hercules, CA, USA), primers, ultrapure water, and DNA. Forty real-time PCR cycles consisted of an initial denaturation at 98 °C for 3 min, denaturation at 98 °C for 15 s, annealing at 51 °C for 30 s (for nuclear DNA) or 47 °C for 30 s (for mitochondrial DNA), and extension at 60 °C for 30 s. A melting curve was conducted at the end of the program to prove the specificity of the PCR products. The PCR was conducted on a qTOWER^3^ real-time PCR cycler (Analytik Jena, Jena, Germany). PCR efficiency was above 95% for both PCR assays. Technical variability on average was below 10%.

### 2.3. DNA Fragmentation

Due to low concentrations of isolated ecDNA, the samples were pooled into two pools, which were concentrated using an Eppendorf centrifuge concentrator plus (Eppendorf, Hamburg, Germany). DNA fragmentation was assessed for the aliquots of centrifuged and DNase I-treated saliva via capillary electrophoresis. A 2100 Bioanalyzer instrument and Agilent High Sensitivity DNA Kit (Agilent, Santa Clara, CA, USA) were used to assess the DNA fragmentation profile. The measurement was done based on the protocol from the manufacturer.

### 2.4. DNase Activity

DNase activity in saliva was measured using a single-radial enzyme-diffusion assay. Briefly, 100 mL agarose gels containing isolated DNA (3.5 mg), fluorescent dye, 2 mM MgCl_2_, 2 mM CaCl_2_, 20 mM Tris HCl pH = 7.5 were prepared and poured. After the gels solidified, small wells were punctured, and samples were pipetted into the wells. Gels were incubated at 37 °C for 16 h in the dark. After incubation, pictures of the gels were made using UVP iBox (Jena, Germany, Analytik Jena), and the diameters of the circles were measured using ImageJ software 1.54 (NIH, Bethesda, MD, USA). DNase activity was calculated based on the standard two-fold serial dilutions of the RNase-free DNase set (Qiagen, Hilden, Germany).

### 2.5. Statistical Analysis

Statistical analysis was conducted in GraphPad Prism 10.1.0. (GraphPad Software. Boston, MA, USA). Two-way ANOVA was used for comparison between saliva centrifuged once or twice and fresh or frozen saliva. The uncorrected Fisher’s LSD test was used as the post hoc test. For correlations, Pearson’s r with *p*-values were reported. *p*-values less than 0.05 were considered statistically significant.

## 3. Results

The concentration of ecDNA in saliva was affected mainly by centrifugation. The average concentration of total ecDNA isolated form fresh saliva centrifuged at 1600× *g* was 1895 ng/mL. Only 2.25% of this ecDNA was still found in the saliva supernatant after the second centrifugation at 16,000× *g*. Similarly, in frozen saliva, the concentration of total ecDNA in 1600× *g* saliva was 2211 ng/mL, and only 1.65% of that ecDNA was still present in the saliva supernatant after centrifugation (Figure 2A, two-way ANOVA: centrifugation F = 15.93, *p* < 0.001; freezing F = 0.09, *p* = 0.76; interaction F = 0.10, *p* = 0.75). A similar pattern was observed for both subcellular fractions. The nuclear ecDNA concentration was lower in samples centrifuged at both 1600× *g* and 16,000× *g* (Figure 2B, two-way ANOVA mixed-effect model: centrifugation F = 14.82, *p* < 0.001; freezing F = 0.71, *p* = 0.41; interaction F = 0.32, *p* = 0.57). Also, mitochondrial ecDNA was affected only by centrifugation (Figure 2C, two-way ANOVA: centrifugation F = 25.79, *p* < 0.001; freezing F = 0.02, *p* = 0.90; interaction F = 0.05, *p* = 0.83). A comparison of the ratio between the mitochondrial genome copy number and the nuclear genome copy number did not show any significant differences caused by either centrifugation or the freezing of saliva. A difference was reported for the interaction of centrifugation and freezing, with a post hoc test showing no differences between groups (Figure 2D, two-way ANOVA mixed-effect model: centrifugation F = 0.36, *p* = 0.55; freezing F = 0.27, *p* = 0.61; interaction F = 4.10, *p* = 0.05). DNase activity in saliva was, on average, decreased by 36% if the samples were frozen before measurement (Figure 2E, *t*-test, t = 4.80, *p* = 0.005).

Correlation analysis was conducted to assess the effect of sample freezing. The correlation matrix for analyzed parameters is summarized in Table 1. The concentration of total ecDNA had a positive correlation between fresh and frozen saliva (Figure 3A, Pearson’s r = 0.58, *p* = 0.02). Similarly, a strong positive correlation between fresh and frozen salivary ecDNA was observed for the nuclear fraction (Figure 3B, Pearson’s r = 0.90, *p* < 0.001) as well as for the mitochondrial fraction (Figure 3C, Pearson’s r = 0.78, *p* < 0.001). However, saliva centrifuged at an additional 16,000× *g* showed no correlation between fresh and frozen aliquots for total ecDNA (Figure 3D, Pearson’s r = 0.23, *p* = 0.44), nuclear ecDNA (Figure 3E, Pearson’s r = 0.15, *p* = 0.66) and mitochondrial ecDNA (Figure 3F, Pearson’s r = 0.46, *p* = 0.10).

The treatment of saliva sample aliquots with DNase I significantly decreased the concentration of total ecDNA in saliva centrifuged at 16,000× *g*, suggesting the presence of free DNA. DNase I removed, on average, 80% of ecDNA in saliva. No effect of DNase I treatment was observed in saliva centrifuged at 16,000× *g* and microparticles. However, total salivary ecDNA was decreased also due to second centrifugation at 16,000× *g* (Figure 4A, two-way ANOVA: centrifugation F = 4.09, *p* = 0.04; DNase I F = 6.04, *p* = 0.04; interaction F = 1.18, *p* = 0.19). On the other hand, no differences caused by centrifugation or DNase I treatment were observed in ncDNA concentration (Figure 4B, two-way ANOVA: centrifugation F = 1.46, *p* = 0.30; DNase I F = 3.35, *p* = 0.10; interaction F = 0.33, *p* = 0.73). This was likely caused by the low number of samples. Several samples yielded no ecDNA after being treated with DNase I and after second centrifugation. Despite the low concertation of ncDNA in the samples, concentrations of mtDNA samples were higher and affected only by centrifugations. The concentration of mtDNA in aliquot centrifuged a second time at 16,000× *g* was significantly lower than in saliva centrifuged only once. This mtDNA removed by second centrifugation represented almost 97%, and it was likely found in microparticles where its concentration was higher than in 16,000× *g* centrifuged saliva (Figure 4C, two-way ANOVA: centrifugation F = 9.54, *p* = 0.002; DNase I F = 2.67, *p* = 0.14; interaction F = 1.51, *p* = 0.25). The concentration of mtDNA in microparticles was 20-fold higher than in 16,000× *g* centrifuged saliva. No effect of DNase I treatment on mtDNA was observed in any of the analyzed aliquots.

Subsequently, the fragmentation of salivary ecDNA was analyzed using capillary electrophoresis. Saliva centrifuged at 1600× *g* mainly had long fragments that were found at the uppermost limit of the detection of the chip (Figure 5). However, two pools had different fragmentation patterns when it came to saliva centrifuged at the additional 16,000× *g*. In pool 1, almost no ecDNA was detected, while pool 2 had three distinct peaks at 400 bp, 500 bp, and 1000 bp. Same peaks that were found in saliva centrifuged at 1600× *g* were observed in ecDNA isolated from microparticles, suggesting that the remaining ecDNA was either free from particles or found in smaller ones. Treatment with DNase I removed all long fragments found in saliva centrifuged at 1600× *g* in pool 1. On the other hand, in pool 2, long fragments were cleaved, creating peaks similar to the centrifuged saliva of pool 2. Overall, DNase I treatment decreased ecDNA concentration but did not remove specific peaks, suggesting that a part of ecDNA was protected from being cleaved.

## 4. Discussion

Preanalytical factors are crucial for the further use of any biomarker, especially for one that can easily be biased due to low concentration in vivo. Recently, this has been clearly shown for the pre-analytical isolation of ecDNA from saliva where different kits and protocols were compared [20]. We went further and analyzed other pre-analytical factors—centrifugation and freezing. The importance of centrifugation protocols was proved and well documented for blood plasma ecDNA [27]. While the first centrifugation pellets cells, the second disposes of cell debris and has been shown to increase the sensitivity of the detection of tumor-associated mutations [28]. How this second step affects the potential diagnostic value of ecDNA in saliva is not clear. We showed that it considerably decreases its concentration. The majority of ecDNA in saliva seems to be associated with cell debris and microparticles, and, based on the observations of this study, is predominantly of mitochondrial origin. Although, other sources could be assessed in relation to saliva too, such as ecDNA originating from bacteria or food.

Freezing has been shown to increase plasma ecDNA, especially of mitochondrial origin in plasma without additional high-spin centrifugation [29]. This is likely caused by the disruption of mitochondria or other vesicles containing mitochondrial DNA by freezing and thawing. Our data do not show similar effects for saliva. Mitochondrial DNA is highly abundant in saliva [30], but the details about its structure, protection, and localization are unknown. Salivary mitochondrial DNA appears to be protected from being cleaved by DNase I, which suggests that it is likely found in microparticles or protected by other mechanisms that prevent degradation by DNase I. These could be whole mitochondria found in the saliva or mitochondria inside larger debris. The different composition of saliva vs. plasma could be an explanation. Salivary mucins and the sialic acid content likely affect the changes induced by freezing [31]. While many analytes in saliva are affected by freezing and thawing, in line with our results, the salivary microbiome analyzed using salivary DNA has been shown to not be affected by freezing [32]. Unsurprisingly, this was also found for total salivary DNA, where freezing and storage did not cause any major losses of DNA [33].

The subcellular origin of ecDNA can be the nucleus or the mitochondria, and, using targeted quantitative PCR, it is possible to distinguish these two different sources [34]. It is not yet clear what the ratio of the mitochondrial copy number in ecDNA means, but it is already found to be altered in some diseases [35,36]. In the context of saliva, mitochondrial DNA has been studied, especially in head and neck cancer, as a predictive marker [24] but potentially also for the detection of the disease [23]. Interestingly, a recent study has shown that salivary ecDNA of a mitochondrial origin is not related to inflammatory markers and varies greatly between, but also within, individuals [30]. In our study, we also found a relatively high variability of mitochondrial ecDNA in saliva, which is only partially related to nuclear or total DNA.

In our study, we analyzed the fragmentation pattern of the ecDNA that has diagnostic value for plasma ecDNA [37]. However, in saliva, the pattern was different in comparison to the one observed in plasma samples. Saliva samples lacked the nucleosomal peak that is present in plasma samples. Longer fragments were observed in the two pools of salivary ecDNA that we assessed. This likely suggests that ecDNA found in saliva was not degraded and resembled DNA from whole saliva [20,38]. Two distinctly different fragmentation profiles were observed in our saliva pools—one having only one peak of long fragments between 2000 bp to 10,380 bp and the other having multiple peaks at 400 bp, 500 bp, 1000 bp and the long 2000 bp to 10,380 bp fragments. The additional fragments found in pool 2 were likely caused by sample degradation.

Additionally, we assessed the protection of ecDNA in saliva from it having been cleaved by DNase I. DNase I can cleave free DNA [39]. Based on our analyses, ecDNA in saliva is mostly protected from being cleaved. However, in saliva centrifuged once, 80% of ecDNA is likely free since it is accessible to be cleaved. This was not observed for ecDNA of nuclear and mitochondrial origin, which could be caused by the low number of samples and a high variability of ecDNA concentration in this sample subset or of a different origin, likely bacterial or food related.

Our study has limitations. We did not sequence the ecDNA isolated from saliva, but we were also not looking for any mutation, tumor, or fetal DNA that might be present in the saliva from pregnant women or diseased patients [19,20,40]. We used saliva samples from young healthy and non-pregnant volunteers that would not have contained these specific sequences, although contamination from other sources could not be ruled out [41]. We rather focused on the quantity of salivary ecDNA, which might be related to inflammation [42,43], cancer [24,36], or other diseases, but might have implications for other fields such as forensic genetics [44]. We did not believe that the size of the study group would be too small. Any major effect of freezing or centrifugation would be provable on 10 to 15 samples, given the use of the dependent tests in statistical analysis, although, of course, a larger study would be needed to analyze the effects of biological factors causing interindividual and not technical variability.

According to our knowledge, this is the first study focusing on the pre-analytical aspects of ecDNA analysis in saliva. In conclusion, our results indicate that an additional centrifugation, but not freezing, of saliva supernatants decreased ecDNA concentration of both nuclear and mitochondrial origin. This indicates that cell debris, apoptotic bodies, and various microparticles could contain most of the ecDNA in saliva. Future similar studies should focus on saliva samples from patients with periodontitis, oral carcinoma, or other diseases to identify the ideal centrifugation protocol for the quantitative analysis of salivary ecDNA.

## Figures and Tables

**Figure 1 diagnostics-14-00249-f001:**
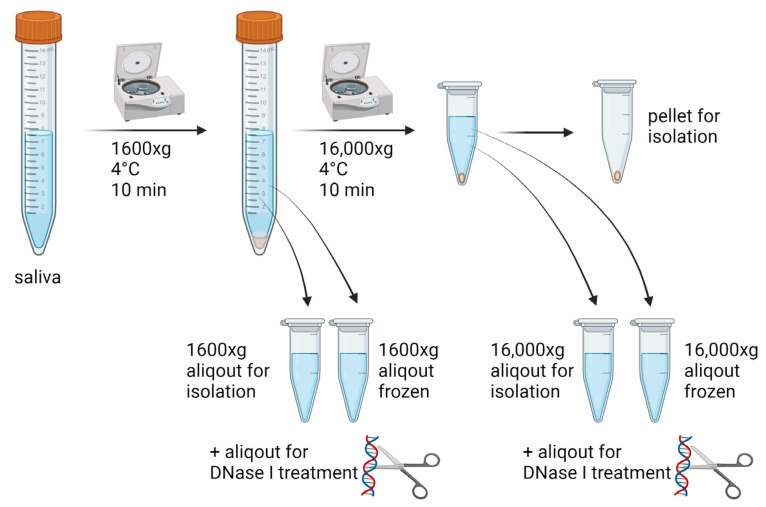
Saliva sample processing, including aliquoting and subsequent variable centrifugation and the freezing of samples before the isolation of extracellular DNA.

**Figure 2 diagnostics-14-00249-f002:**
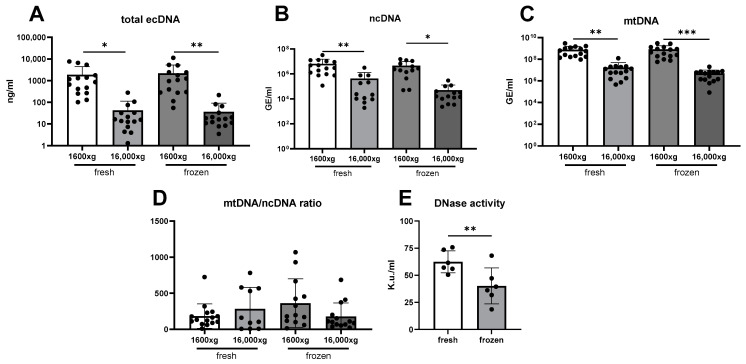
(**A**) Total ecDNA, (**B**) nuclear ecDNA, (**C**) and mitochondrial ecDNA were lower in samples centrifuged additionally at 16,000× *g*. (**D**) The ratio between mitochondrial ecDNA/nuclear ecDNA genome copy number did not differ between the aliquots of saliva. (**E**) DNase activity decreases with the freezing of saliva. ecDNA—extracellular DNA, ncDNA—nuclear ecDNA, mtDNA—mitochondrial ecDNA. *—*p* < 0.05, **—*p* < 0.01, ***—*p* < 0.001.

**Figure 3 diagnostics-14-00249-f003:**
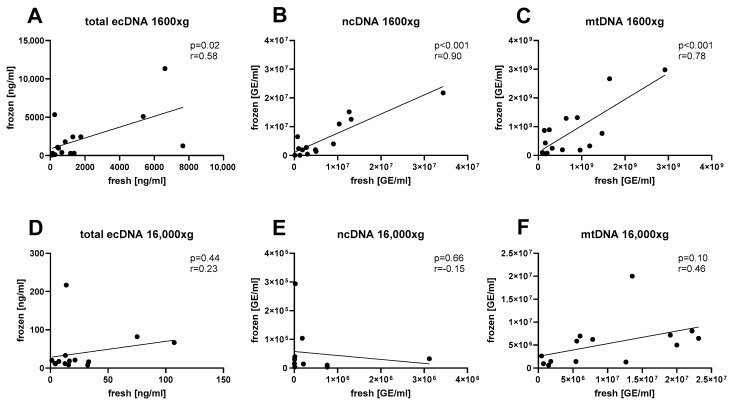
Correlations between (**A**) total ecDNA, (**B**) nuclear ecDNA, and (**C**) mitochondrial ecDNA isolated from fresh and frozen saliva were assessed in aliquots centrifuged at 1600× *g*, as well as in aliquots of saliva centrifuged at 16,000× *g* for the same subsets of (**D**) total ecDNA, (**E**) ncDNA, and (**F**) mtDNA. ecDNA—extracellular DNA, ncDNA—nuclear ecDNA, mtDNA—mitochondrial ecDNA.

**Figure 4 diagnostics-14-00249-f004:**
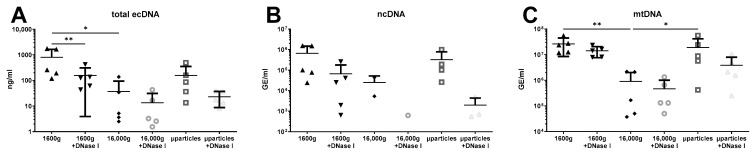
Saliva of healthy individuals contains protected (**A**) total ecDNA, (**B**) ncDNA, and (**C**) mtDNA. ecDNA—extracellular DNA, ncDNA—nuclear ecDNA, mtDNA—mitochondrial ecDNA, µparticles—microparticles. *—*p* < 0.05, **—*p* < 0.01.

**Figure 5 diagnostics-14-00249-f005:**
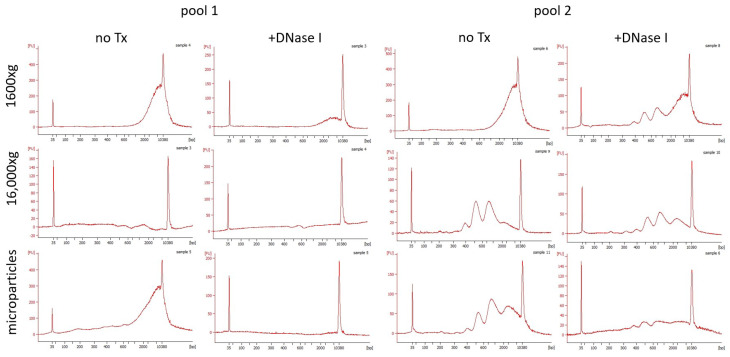
Fragmentation of pooled ecDNA in saliva aliquots assessed on capillary electrophoresis. No Tx—no treatment.

**Table 1 diagnostics-14-00249-t001:** Correlation matrix of salivary ecDNA concentrations with freezing and centrifugation protocols as potential factors. ecDNA—extracellular DNA, ncDNA—nuclear ecDNA, mtDNA—mitochondrial ecDNA, *—*p* < 0.05, **—*p* < 0.01, ***—*p* < 0.001.

			Fresh						Frozen					
			1600× *g*			16,000× *g*			1600× *g*			16,000× *g*		
			Total ecDNA	ncDNA	mtDNA	Total ecDNA	ncDNA	mtDNA	Total ecDNA	ncDNA	mtDNA	Total ecDNA	ncDNA	mtDNA
fresh	1600× *g*	total ecDNA												

		ncDNA	0.89											
			***											
		mtDNA	0.68	0.61										
			**	*										
	16,000× *g*	total ecDNA	0.48	0.64	0.68									
				**	**									
		ncDNA	−0.10	−0.07	0.16	0.06								

		mtDNA	0.47	0.63	0.68	1.00	0.20							
				**	**	***								
frozen	1600× *g*	total ecDNA	0.64	0.48	0.76	0.32	−0.14	0.32						
			**		***									
		ncDNA	0.86	0.90	0.81	0.67	−0.03	0.67	0.84					
			***	***	***	**		**	***					
		mtDNA	0.45	0.40	0.78	0.35	−0.16	0.35	0.79	0.62				
					***				***	*				
	16,000× *g*	total ecDNA	0.51	0.66	0.71	1.00	−0.09	0.99	0.36	0.70	0.39			
			*	**	**	***		***		**				
		ncDNA	0.50	0.65	0.70	1.00	−0.15	0.99	0.34	0.69	0.37	1.00		
				**	**	***		***		**		***		
		mtDNA	0.49	0.65	0.69	1.00	−0.09	0.99	0.34	0.69	0.37	1.00	1.00	
				**	**	***		***		**		***	***	

## Data Availability

The data obtained from this study are available from the authors upon reasonable request.

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
