# Peer review of "Pre-Analytical Factors Affecting Extracellular DNA in Saliva"

_diagnostics, 2024, doi:10.3390/diagnostics14030249_

Round 1
Reviewer 1 Report
Comments and Suggestions for Authors
After reading the manuscript "Pre-analytical factors affecting extracellular DNA in saliva" I believe that the research topic is well-aligned with the scope of the journal and is relevant to current research interests.
In my view, there is currently a shortage of methodological research. The presented study is a successful attempt to fill this gap.
I suggest making a minor revision berfore acceptance. Below are just a few comments to improve the text of the manuscript
1) Line 81. It should be clarified how the volunteers’ oral health was confirmed
2) Line 136 and Fig. 1. 4C and 37C needs to be replaced by 4℃ and 37℃
3) Please, consider to change the format of Table 1. It is difficult to understand in the present form.
Author Response
Response to reviewer 1
We are grateful to reviewer 1 for his/her constructive and positive evaluation. Here is our point by point response.
Reviewer 1:
After reading the manuscript "Pre-analytical factors affecting extracellular DNA in saliva" I believe that the research topic is well-aligned with the scope of the journal and is relevant to current research interests.
Our response:
We agree with the reviewer. We and others are interested in the non-invasive diagnostics and the huge potential of extracellular or cell-free DNA in various body fluids. However, the technical aspects need to be studied in detail to prevent bias in further large clinical studies.
Reviewer 1:
In my view, there is currently a shortage of methodological research. The presented study is a successful attempt to fill this gap.
Our response:
Exactly that was our primary motivation to start this study. While more and more papers on the use of salivary DNA are published, the basic methodology is not clear yet and should be tested in advance.
Reviewer 1:
I suggest making a minor revision berfore acceptance. Below are just a few comments to improve the text of the manuscript
Our response:
We are grateful for these constructive suggestions. We have tried to follow all of them and hope that the improved version of the manuscript is now clearer.
Reviewer 1:
1) Line 81. It should be clarified how the volunteers’ oral health was confirmed
Our response:
The reviewer is right that the oral health should have been studied in detail. However, we only have a negative outcome from the last dental preventive check-up. No specific examinations have been conducted as part of this study. On the other hand, this shortcoming might actually be a strength as well, as the outcomes might be valid on a relatively variable population.
Reviewer 1:
2) Line 136 and Fig. 1. 4C and 37C needs to be replaced by 4℃ and 37℃
Our response:
Thank You for pointing to these misspellings, we have corrected these errors.
Reviewer 1:
3) Please, consider to change the format of Table 1. It is difficult to understand in the present form.
Our response:
We have changed the format of the table, so that it fits the page format. Hopefully, this will not be changed again during the conversion of the file.
Reviewer 2 Report
Comments and Suggestions for Authors
Related to “Pre-analytical factors affecting extracellular DNA in saliva”
# Comments
DNA in physiological fluids are important in diagnosis approach. So, in my opinion this manuscript has good potential for publication in “Diagnostic” journal.
The paper has good quality and recommends for publication after minor revision
v Please up-date old references (3, 7, 31,41 and etc.).
v The novelty of the study must be highlighted in the abstract and conclusion sections.
v Please extend the figure.2 captions’.
v I recommend the highlighting of advantages and disadvantages of “developed method in this study and compare with other published methods” Presentation in a table is highly appreciated.
v Minor editing of English language required.
Comments on the Quality of English Language
Minor editing of English language required.
Author Response
Response to reviewer 2
We are grateful to reviewer 2 for his/her constructive and positive evaluation. Here is our point by point response.
Reviewer 2:
DNA in physiological fluids are important in diagnosis approach. So, in my opinion this manuscript has good potential for publication in “Diagnostic” journal.
Our response:
We agree with the reviewer 2. The diagnostic value of any biomarker is dependent on the knowledge of its physiology and the technical aspects of its variability. This should be a contribution to this issue for salivary DNA.
Reviewer 2:
The paper has good quality and recommends for publication after minor revision
Our response:
We thank the reviewer 2 for the positive evaluation and for the highly constructive comments to our manuscript.
Reviewer 2:
v Please up-date old references (3, 7, 31,41 and etc.).
Our response:
We agree with the reviewer 2 that new references related to the topic should be used. We have cited several papers published during the last year. The mentioned citations, however, are crucial papers that have demonstrated some phenomena for the first time. The first description of the neutrophil extracellular traps for example must be cited in such a manuscript. The existence of the transrenal DNA or the effects of freezing and thawing are papers that cannot be simply replaced by some new ones. Number 41 is a relevant paper from our group and it would be not appropriate to ignore our own relevant and previously published papers. That is why we suggest to leave these citations and rather focus on the latest as well.
Reviewer 2:
v The novelty of the study must be highlighted in the abstract and conclusion sections.
Our response:
We have added sentences highlighting the novelty of the study as suggested by Reviewer 2.
Reviewer 2:
v Please extend the figure.2 captions’.
Our response:
We have increased the size of the Figure as well as the text used within.
Reviewer 2:
v I recommend the highlighting of advantages and disadvantages of “developed method in this study and compare with other published methods” Presentation in a table is highly appreciated.
Our response:
We agree with the reviewer that a table would be ideal if there are systematic comparisons possible. As we see the current status, there are very little systematic data available. In other words, there is nothing to compare yet. That is actually the rationale for our study and so, we hope that this will be understood by the readers as well as by the reviewers.
Reviewer 2:
v Minor editing of English language required.
Our response:
We have polished the text and hope that it is now improved.